# Distinctive features of lipoprotein profiles in stroke patients

**Tomokazu Konishi** [1]*, **Yurie Hayashi**[1], **Risako Fujiwara**[2], **Shinpei Kawata**[3], **Tatsuya Ishikawa**[3]

**1** Graduate School of Bioresource Sciences, Akita Prefectural University, Akita, Japan, **2** Cardiovascular Internal Medicine, Akita City Hospital, Akita, Japan, **3** Research Institute of Akita Cerebrospinal and Cardiovascular Center, Akita Prefectural Hospital Organization, Akita, Japan

* konishi@akita-pu.ac.jp

**Data Availability Statement:** All relevant data are within the paper and its Supporting Information files.

## Abstract

Classes of lipoproteins solubilize lipids in the blood, and their profiles are important for preventing atherosclerotic diseases. These can be identified by gel filtration HPLC, which has been analyzed in a manner that yields the same values as the *de facto* standard method, i.e., ultracentrifugation; however, previous studies have found that ultracentrifugation and its simplified alternatives, enzymatic methods, yield incorrect values. Here HPLC data of stroke patients and the controls were compared using data-driven analyses, without consideration for ultracentrifugation. The data well-separated patients from controls. In many patients, the level of HDL1 (a cholesterol scavenger) was low. The TG/cholesterol ratio of chylomicrons was found to be low in patients and high in the healthy elderly; the lower level may indicate a larger intake of animal fats. High levels of free glycerol in the elderly were hazardous, suggesting more dependence on lipids as an energy source. Statins had minimal effect on these factors. LDL cholesterol, the commonly-used risk indicator, was not a risk factor actually. Enzymatic methods failed to separate the patients from the control; hence, the existing guidelines for screening methods and medical treatment need to be revised. As an immediate step, glycerol would be an adaptable indicator.

## Introduction

For many years, lipoprotein classes have been separated using isopycnic ultracentrifugation [1–5], and the quantification of the classes was based on this separation. Ultracentrifugation, however, is a time-consuming and expensive method that it is not suitable for examining large numbers of samples during medical screening. Simple kits have been developed to overcome this situation that are expected to solubilize one of the classes with specific surfactants; the solved cholesterol is determined through enzymatic methods [6]. These results reproduced the results of ultracentrifugation accurately. Newer methods such as high-performance liquid chromatograph (HPLC) and nuclear magnetic resonance interpret data by assigning peak signals to one of the lipoprotein classes in a process that was analyzed to coincide with ultracentrifugation results [7, 8]. Of course, this is because ultracentrifugation is the gold standard.

When HPLC data were analyzed in a data-driven manner and samples were divided into fractions for biochemical examinations, the results were significantly different than those of

**Funding:** The author(s) received no specific funding for this work.

**Competing interests:** The authors have declared that no competing interests exist.

previous interpretations [9]. For example, high-density lipoproteins (HDL) were much more minor than previously believed, while all classes of lipoproteins contained far more proteins than previously thought [10]. Ultracentrifugation uses a large gravitational force onto a sample over a long period of time. While this is a good method for single molecules such as DNA, lipoproteins are inevitably fragile complexes wherein proteins and lipids are associated with hydrophobic bonds that could break down, which could have been actually occurring. This phenomenon was demonstrated in experiments with rats but was later found to occur in human samples as well [11]. This error naturally affects the enzymatic methods and the measured low-density lipoproteins (LDL) and HDL values are mixtures of several classes [11].

This error raises the possibility that the measured values do not indicate medical conditions, although both LDL and HDL are factors that are expected to represent pathology. This may be the reason why LDL cholesterol is not always a significant risk factor, even in large cohort studies [12, 13]. Nevertheless, LDL cholesterol measured using the enzymatic method is used as a guideline for existing medical treatments [14]. When it is high, statins (i.e., inhibitors of cholesterol synthesis) are prescribed to prevent strokes.

This study compared serum samples of patients who had suffered from untreated strokes to samples from healthy controls, using HPLC; HPLC data were analyzed in a data-driven manner. The quantity of some classes of lipoproteins was a clear indicator of the differences between the groups, although the current methods failed to find any of these differences.

## Materials and methods

For the experimental procedures and conditions, see our previous article on the younger volunteers under 60 ($n = 59$) [11]. As these volunteers were somewhat younger than the patients, we recruited $n = 11$ new older volunteers (see S1 Fig in S1 File). For patients, we used data from people who were transported to the emergency room in 2018 at the Akita Cerebrospinal and Cardiovascular Center in Akita, Japan ($n = 189$) (see S1 Table in S1 File for individual symptom records; many were cerebral infarctions, but some have cardiac symptoms). The study was approval from the ethics committee of Akita Cerebrospinal and Cardiovascular Center (ID. 19–21); participants were given informed consent and written consent forms were collected in a document form. All data were anonymized prior to analysis. Participants did not include minors.

Triglycerides (TG) and cholesterol levels were monitored over time for HPLC elution, and the amount of each class was estimated by curve-fitting the data with minimal normal distribution parameters [9]. Data were compiled using principal component analysis (PCA) (see S2 Fig in S1 File; raw data and the results are shown in S2–S4 Tables in S1 File) [15]. Chylomicron (CM)1 and CM2 were removed from the PCA because we could not specify the time after meals, and their values thus fluctuated. The peaks of LDL2 and LAC2 overlapped, and they were added together since they could not always be reproducibly separated. Cholesterol for TG-rich VLDL (TR) was omitted because it was too low to measure.

Logarithms were taken for the amount of each class measured by HLPC or the enzymatic methods, as they are lognormally distributed [11]. For some classes that were confirmed to be significant, the log ratio of TG/cholesterol was also included; it was important to include CM in the analysis. Free glycerol was calculated by multiplying the amount of glycerol measured as TG by 31.2/300, which is the inverse of the estimation of TG measured by its degradant, glycerol. These were subtracted by the robust (i.e., trimmed) mean and divided by the robust standard deviation (i.e., median absolute deviation) of the young control group to calculate the z-score.

Z-scores thus obtained were each approximately normally distributed; however, in some cases when the amount of a class was very small, the curve fit failed and ignored the residuals, producing a negative z-score as an outlier. For cases in which the z-score was < -4, it was replaced by -4, which was the lower limit at which the distribution broke down. PCA [15] was calculated using these z-scores.

## Results

The results of PCA for each dataset in this research are displayed in a biplot that shows the PCs of the sample and the item simultaneously (see Fig 1; lower PC data are shown in S3 Fig in S1 File, while raw data and PCA results are shown in S2–S4 Tables in S1 File). The data is obtained as a sample × item (quantity of each class) matrix; PCA finds common data movements within that matrix and sets axes to represent them. Each axis gives a unique PC for the sample and item. Since the PCs of the samples and items are complementary, it is clear which item affected which sample. Dots and single letters show PCs for samples—red dots indicate controls, blue dots indicate older controls, black letters indicate strokes, and 's' indicates taking statins. The results of HPLC clearly separated controls from patients, although older patients had larger values in PC1 than the younger ones (see Fig 1A and S3 Fig in S1 File). In contrast, the conventional method resulted in virtually no separation, although patients and controls might have slightly different distributions (see Fig 1B and S4 Fig in S1 File).

PCs of the items indicate the classes that contributed PCs for samples (see Fig 1, class names; only representative ones are shown). In Fig 1A, orange letters represent TG of the classes, blue letters represent cholesterol, and green letters represent the TG/cholesterol ratio; typical examples are shown as box plots (see Fig 2). PC1 was formed by reduced HDL-1 as well as increased LDL-antiprotease-complex (LAC)1 [9] and TG of LDL1 in patients (see Figs 1A, 2A and 2B). Lp(a), a particularly cholesterol-rich class of VLDL [11], was shown to be a factor in PC1 (see Fig 1A). Lp(a) is known as a risk factor [16].

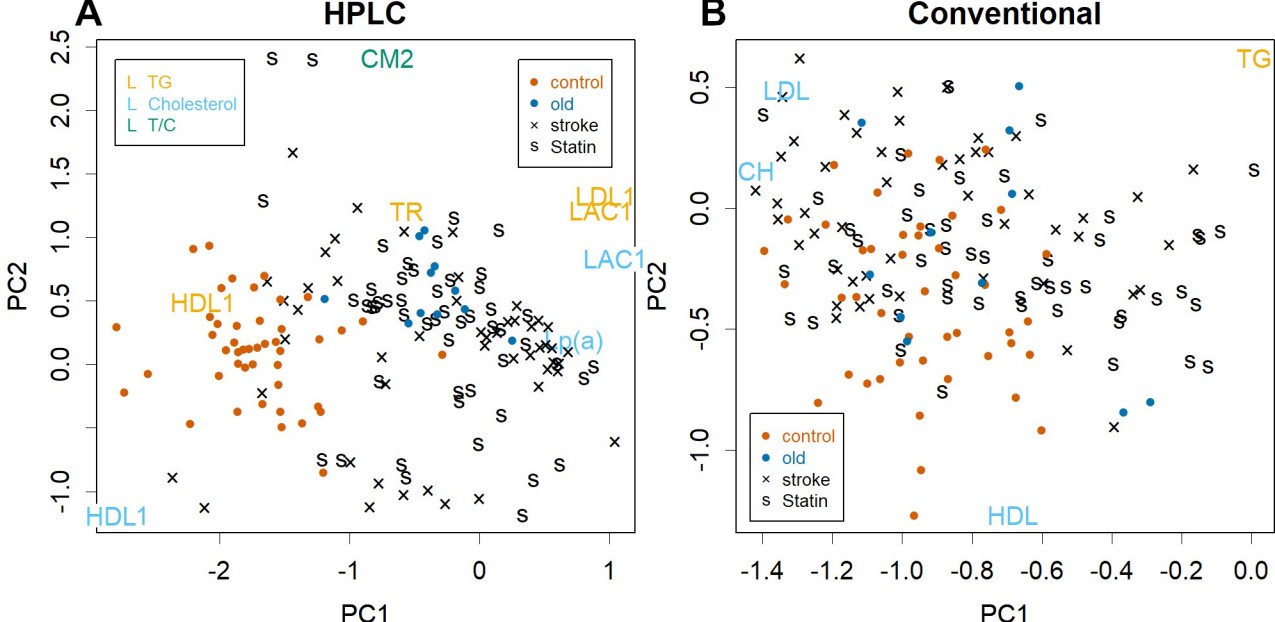

**Fig 1. Biplot display of PCA. A**. amount of each class and **B**. conventional enzymatic method (commonly used). Red dots: control samples; blue dots: elder controls, black: stroke patients, s; statin takers. In panel A, class names represent PC for representative items. Orange: TG, blue: cholesterol, green: TG/cholesterol ratio. In panel B, TG: total TG, CH: total cholesterol. In the panels, axes are for items; those for samples are omitted for simplicity (see S3 Fig in S1 File).

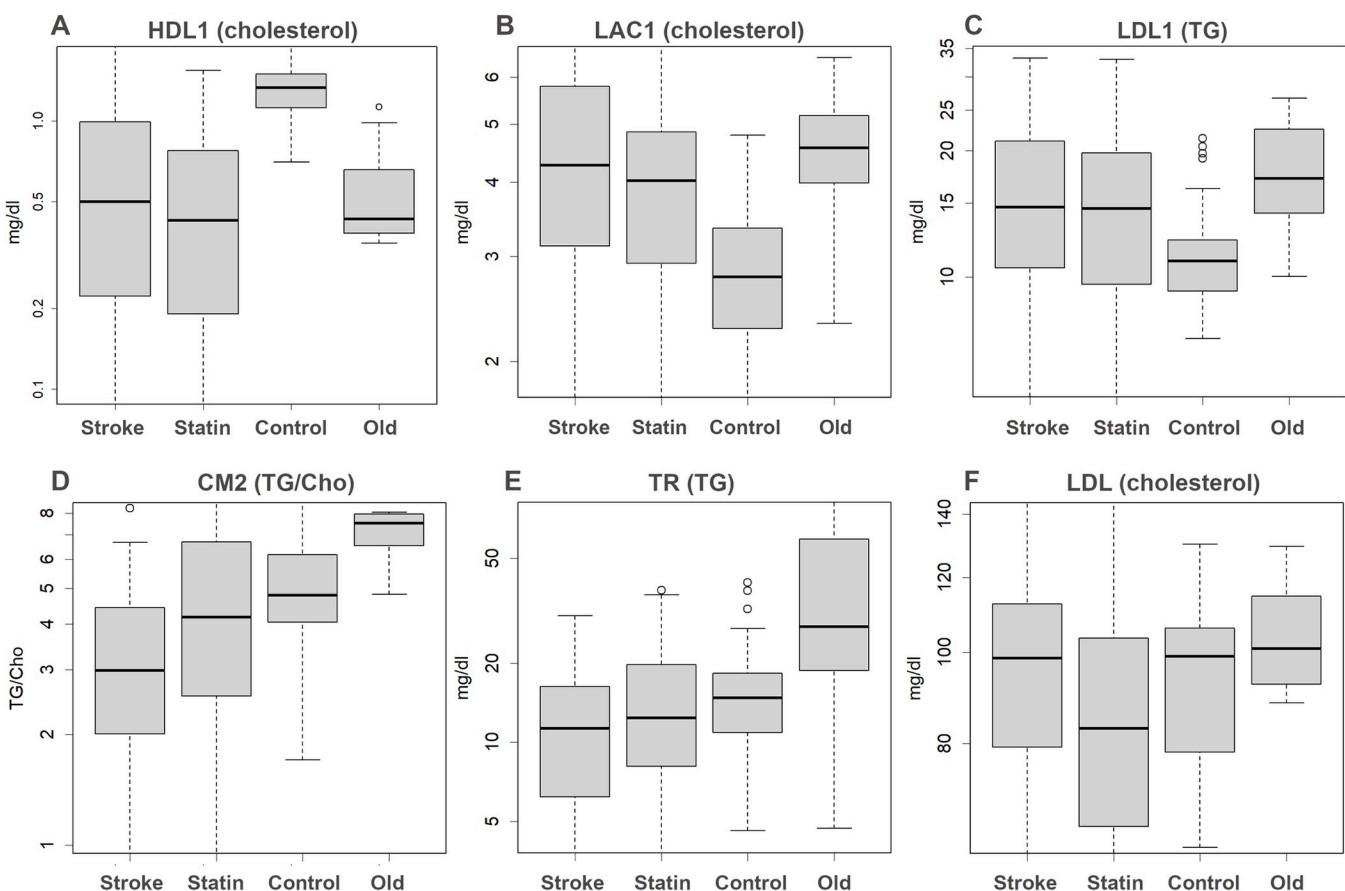

**Fig 2. Boxplots.** Each of these is an amount or a ratio for the class indicated. **A**. Largest difference was observed in HDL1 cholesterol, which was decreased in the patients, **B** and **C**. Higher LAC1 cholesterol and TG of LDL1 are risks, **D**. TG/cholesterol ratio of CM, showing this ratio in the diet, **E**. TR (TG-rich VLDL), **F**. LDL cholesterol, which is the target of statins. The y-axis is shown logarithmically.

Differences were observed also in lower PCs with respect to HPLC results. For example, the TG/cholesterol ratio of CM2 is presented in PC2 (see Figs 1A and 2D); this ratio was lower in patients and higher in healthy elderly individuals. Reduced TR, which is mostly made up of TG [11], may also be a risk factor (see Figs 1A and 2E)—this may indicate that patients depend on TG levels from food and not from the liver; in contrast, healthy elderly individuals have higher TR levels. Even among lower PCs, those of some patients were more extreme than controls (see S3 Fig in S1 File). This indicates that each patient had abnormal values for particular classes, which were not shown in PC1 and PC2, since the trend was different from that of HDL1/LAC1 and CM & TR. Each patient had a different pathology and since these phenomena occur only in a subset of the population, statistical tests between patients and controls would not necessarily always yield significant differences (see S5 Fig in S1 File).

The existing method failed to separate patients from controls because each measure differed little between the classes (see Fig 1B, S4 and S6 Figs in S1 File). This is due to the fact that HDL and LDL measured by existing methods do not detect the classes for which they are intended but instead measure a mixture of several classes [11]. Since such classes move separately, their summation becomes an ineffective indicator.

Free glycerol is a test that is rarely used for health screening but it showed certain PC values relevant to this research (see S3 Fig in S1 File). It can thus be a good indicator; in fact, patients and controls were well separated on a scatter plot of age vs. glycerol (see Fig 3A)—0.6 mg/dL

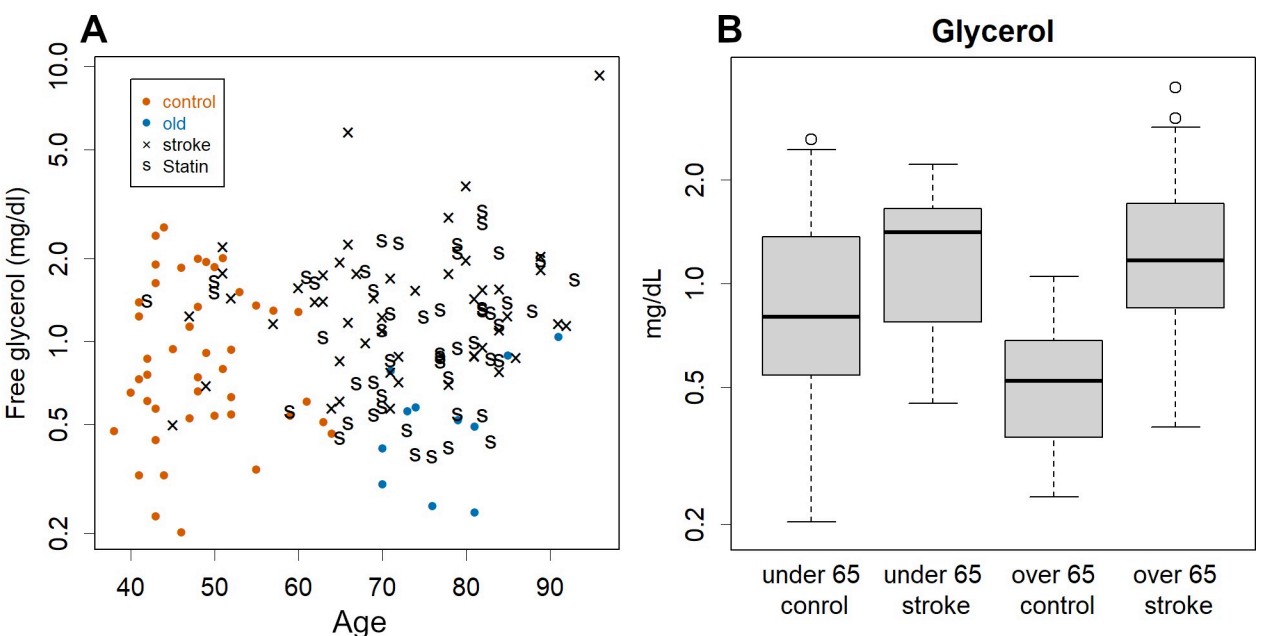

**Fig 3. Glycerol levels and age. A**. scatter plot; note the good separation between patients and controls. **B**. Box plot of free glycerol. There is a clear difference at least for those over 65 years of age. Free glycerol presumably depends on the rate of TG degradation, which occurs while storing or consuming fatty acids; the high level suggests that they rely much on lipids as an energy source.

glycerol in individuals over the age of 65 is considered dangerous (see Fig 3B and S5 Fig in S1 File).

Due to the limited data in this study, we have not considered a threshold value for each lipoprotein. However, for reference, considering the safe lower limit for HDL1 as 1 mg/dl, the relative risk of falling below that threshold was 2.9, with an odds ratio of 16. Taking the safe upper limit for LAC1 as 4 mg/dl, the relative risk of exceeding it was 1.8, with an odds ratio of 5.6. Similarly, the relative risk was 1.3 and the odds ratio was 13 for subjects over 65 years of age with free glycerol levels greater than 1 mg/dl.

Statins did not show much difference in the amount of each class in PCA (see Fig 1A and S3 Fig in S1 File). In fact, they lowered cholesterol in Lp(a) and LDL-1; however, although Lp(a) would be a risk factor, LDL was not a factor that separated controls from patients. The level in stroke patients was found to be the same as that in the experimental controls (see Fig 2F and S5 Fig in S1 File).

## Discussion

This was a retrospective study to assess the relationship between lipoproteins and pathophysiology in a short period of time by comparing stroke patients with controls and thus has certain limitations. Whether the factors presented in this study can actually be used for diagnoses should be validated and supported by prospective cohort studies; for example, it is not impossible for the lipoprotein profile to change rapidly after stroke. Nevertheless, such a change would require large alterations in synthesis or degradation, which is very unlikely. These factors may produce better results than adhering to the current guideline, which has been negated herein due to its ineffectiveness (see Fig 1B).

The most frequently observed risk factor found by HPLC was a decrease in HDL1 levels (see Figs 1A and 2A). This means that the scavenger was reduced, disturbing cholesterol

recovery. Similarly, increased LAC1 and TG levels in LDL1 were risk factors; these phenomena were observed in many patients but it is noteworthy that they did not always occur at the same time and not all patients had them. Another type of patient had extreme values outside PC1 with different tendencies in the data. Interestingly, older volunteers had lower HDL1 and higher LAC1 and LDL1 TG (see Fig 2A–2C), with higher values for PC1 (see Fig 1A). Ageing is another risk factor [14].

Statins did not show clear differences in PCA (see Fig 1, S3 and S5 Figs in S1 File). Although it is possible that statin-ineffective patients experienced a stroke, as cholesterol was actually lowered in Lp(a) and LDL1 (see Fig 2F and S5 Fig in S1 File), this worked as an inhibitor of cholesterol synthesis. Statins had no effect on the other factors that separated patients from controls, as expected from the principle; this limited effect may not be sufficient to prevent a stroke.

Existing enzymatic methods failed to separate patients—the currently used index [14] was irrelevant and needs to be revised. If HPLC is always available, it would be ideal; however, this is also a labor-intensive method that is not suitable for large numbers of samples. Therefore, the conventional method for measuring HDL and LDL levels should be improved immediately. If HDL1 and/or LAC1 can be measured, they can be used as primary indicators of danger. Similarly, if only LDL1 can be solubilized, TG would also be a good indicator. There are several new reports on the risk of LDL cholesterol, for example [17–20], but it would surely be a clearer conclusion if the factors presented here were examined. As shown in Fig 1B, patient and control groups were distributed slightly differently even in the conventional methods. In a large cohort study, the significance would be observed even with such faint differences; however, such statistical significance does not guarantee their importance.

How do the elderly volunteers adjust to their age? The effects of aging were observed at higher PC1 values (see Fig 1A) but they exhibit some other remarkable characteristics. One was the high ratio of TG in CM (see Figs 1A and 2D)—as CM is the primary particle produced from food in the intestine, it directly reflects ingredient uptake. Hence, a high ratio of TG means that the cholesterol they ingested was less; they seemed to avoid animal diets. The opposite was observed in the patients (see Fig 2D)—cholesterol is synthesized in the liver anyway, but the irregular and excessive intake of cholesterol may increase the risk of the same even further. TR in healthy elderly people was higher (see Fig 2E), probably because the necessary TG was supplied by the liver rather than by the diet and since it was not consumed as much. The opposite was again observed in the patients.

Fortunately, free glycerol represented the risk well, clearly separating patients (see Fig 3). Glycerol can be measured easily, which is why these values should be used proactively to identify elderly people at potential risk. Unfortunately, the separation was not sufficient in younger people; therefore, it is desirable to have kits for HDL1, LAC1, and LDL-TG.

Low glycerol levels probably indicate lower dependence on lipids as fuel. The level of a substance is determined by the balance between the addition to and subtraction from the system. Glycogenesis in the liver is the main subtraction of glycerol. This is on a smaller scale than sugar intake from diet and is probably not strictly regulated. The main addition here is the breakdown of TG in adipocytes or lipoproteins. This increases when lipids are consumed or stored, which is important for homeostasis and regulation of energy balance. It is likely that this addition determines glycerol levels.

This study was conducted in a small area in Japan. Hence, there may not be many variations in the lifestyles or genetic background of the population. For example, because obesity is more prone to obesity-induced diabetes [21, 22], extreme obesity is rare in Japanese individuals [23]. It is unexpected that LDL cholesterol was not found to be a risk factor in this study but a more diverse population may result in different conclusions.

## Conclusion

Although this was a retrospective study that compared stroke patients with controls, the factors found may indicate the conditions and therefore should be targeted in future prospective cohort studies. The most frequently observed risk factor was decreased HDL1, followed by increased LAC1 and TG levels in LDL1; statins did not appear to be effective against these factors. The enzymatic methods failed to separate patients, and the currently used guidelines for medical treatment need to be re-examined. Although healthy elderly volunteers showed some signs of risk, they had other remarkable characteristics: TG-rich CM and higher TR. This suggests that there was less meat and sugars in the diet. Free glycerol represented the risk well, clearly separating patients. Glycerol can be measured easily and should thus be used to identify people at potential risk.

## Supporting information

**S1 File.**
(ZIP)

## Acknowledgments

We would like to thank Editage (www.editage.com) for English language editing.

## Author Contributions

**Conceptualization:** Tomokazu Konishi.

**Data curation:** Tomokazu Konishi, Shinpei Kawata.

**Formal analysis:** Tomokazu Konishi.

**Investigation:** Yurie Hayashi, Risako Fujiwara.

**Methodology:** Tomokazu Konishi.

**Project administration:** Tomokazu Konishi, Risako Fujiwara, Shinpei Kawata, Tatsuya Ishikawa.

**Software:** Tomokazu Konishi.

**Supervision:** Tatsuya Ishikawa.

**Validation:** Tomokazu Konishi.

**Visualization:** Tomokazu Konishi.

**Writing – original draft:** Tomokazu Konishi.

**Writing – review & editing:** Tomokazu Konishi.

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
