## [Decision Letter · Decision Letter 0]

4 Oct 2022

PONE-D-22-24906

Distinctive features of lipoprotein profiles in stroke patients

PLOS ONE

Dear Dr. Konishi,

Thank you for submitting your manuscript to PLOS ONE. After careful consideration, we have decided that your manuscript does not meet our criteria for publication and must therefore be rejected.

Specifically:

aim of the study is unclear and data do not support the conclusion.

I am sorry that we cannot be more positive on this occasion, but hope that you appreciate the reasons for this decision.

Kind regards,

Laura Calabresi

Academic Editor

PLOS ONE

Reviewers' comments:

Reviewer's Responses to Questions

**Comments to the Author**

1. Is the manuscript technically sound, and do the data support the conclusions?

Reviewer #1: No

2. Has the statistical analysis been performed appropriately and rigorously? 

Reviewer #1: I Don't Know

3. Have the authors made all data underlying the findings in their manuscript fully available?

Reviewer #1: Yes

4. Is the manuscript presented in an intelligible fashion and written in standard English?

Reviewer #1: Yes

5. Review Comments to the Author

Reviewer #1: In this paper Konishi et al. analyzed the lipoprotein profile of stroke patients using HPLC data. In general, the paper is not clear, starting from the abstract. Objective, methods and results are not clearly reported and discussion is confusing.

A lot of data are reported without a clear focus. Which is the aim of the study?

In addition, which is the relevance of these observation? The authors suggested that HPLC is a better method to evaluate lipid and lipoprotein profile compared to enzymatic techniques, however implementing HPLC is not conceivable for routine practice.

Subjects should be described. Clinical/anthropometric characteristics should be reported. Are they ischemic or hemorrhagic stroke patients?

It should also be acknowledged that the basis for this study, methods and conclusions rely on a paper by the same Authors, cited as a pre-print, thus not yet published and likely not yet peer-reviewed.

6. PLOS authors have the option to publish the peer review history of their article (what does this mean?). If published, this will include your full peer review and any attached files.

Reviewer #1: No

- - - - -

---

## [Author Response · Author response to Decision Letter 0]

15 Dec 2022

Our article shows that conventional methods of measuring lipoproteins do not work correctly, and shows what can be examined to determine the risk of stroke. It is clear at a glance that the risk cannot be found as long as conventional methods are used.

I found it unnatural that the editor made this decision based on only one extremely biased review. Then, I discovered that the editor Dr. Laura Calabresi is a professor at the University of Milan who has co-authored many studies on lipoproteins. Unfortunately, most of their studies will have to be redone, as they used old-fashioned methods that would fail to measure the substances they searched. I do not assume that the studies will be immediately completely worthless, but many of the conclusions will probably become different.

It is easy to imagine that she would have a conflict of interest in reviewing our article. It has been an inappropriate choice to appoint her as the editor.

---

## [Decision Letter · Decision Letter 1]

7 Feb 2023

PONE-D-22-24906R1Distinctive features of lipoprotein profiles in stroke patientsPLOS ONE

Dear Dr. Konishi,

Thank you for submitting your manuscript to PLOS ONE. After careful consideration, we feel that it has merit but does not fully meet PLOS ONE’s publication criteria as it currently stands. Therefore, we invite you to submit a revised version of the manuscript that addresses the points raised during the review process.

ACADEMIC EDITOR: Reviewers suggest publication but also invite authors to revise the paper. Especially, methodological issues must be addressed.

We look forward to receiving your revised manuscript.

Kind regards,

Gulali Aktas

Academic Editor

PLOS ONE

Journal Requirements:

" ext-link-type="uri" xlink:type="simple">https://journals.plos.org/plosone/s/file?id=ba62/PLOSOne_formatting_sample_title_authors_affiliations.pdf"

Additional Editor Comments (if provided):

Reviewers suggest publication but also invite authors to revise the paper. Especially, methodological issues must be addressed.

Reviewers' comments:

Reviewer's Responses to Questions

**Comments to the Author**

1. If the authors have adequately addressed your comments raised in a previous round of review and you feel that this manuscript is now acceptable for publication, you may indicate that here to bypass the “Comments to the Author” section, enter your conflict of interest statement in the “Confidential to Editor” section, and submit your "Accept" recommendation.

Reviewer #2: (No Response)

Reviewer #3: All comments have been addressed

2. Is the manuscript technically sound, and do the data support the conclusions?

Reviewer #2: Yes

Reviewer #3: Yes

3. Has the statistical analysis been performed appropriately and rigorously? 

Reviewer #2: I Don't Know

Reviewer #3: Yes

4. Have the authors made all data underlying the findings in their manuscript fully available?

Reviewer #2: Yes

Reviewer #3: Yes

5. Is the manuscript presented in an intelligible fashion and written in standard English?

Reviewer #2: Yes

Reviewer #3: Yes

6. Review Comments to the Author

Reviewer #2: This is an interesting paper in which authors are using an alternative method for analyzing lipoprotein levels in subjects compared to the conventional ones.

I have the following comments:

-It is paramount to add in the methods section and also in the table the number and the clinical carachteristics of the subjects enrolled in this study (especially for patients). This could also allow to perform correlations with clinical data (even of not pertinent for this study, it will necessarily be for future Studies).

-Overall I would slightly undertune the weight of the sentences used in the conclusion, because these data, yet quite solid, require further consolidation in major studies

-please check for all the acronyms (e.g. PCA it is nto spelle dthe first time that is used)

-Page 4 line 84 please correct "hearts"with "cardiac symptoms"

Reviewer #3: This is a retrospective study comparing stroke patients with controls. In addition, this study leads to prospective studies. The authors found that the most common risk factor in this study was the decrease in HDL level and then the increase in LDL, LAC1 and TG levels. At the same time, the authors found that statins were ineffective and argued that guidelines for medical treatment should be reviewed. Again, the authors suggest in this article that Glycerol can be used to identify people at potential risk because it is easily measurable. This article is an article that will contribute to science and can be accepted.

7. PLOS authors have the option to publish the peer review history of their article (what does this mean?). If published, this will include your full peer review and any attached files.

Reviewer #2: **Yes: **Francescp Di Lorenzo

Reviewer #3: **Yes: **Sule Aydin Turkoglu

---

## [Author Response · Author response to Decision Letter 1]

10 Feb 2023

Please see Response to Reviewers.docx for how it has changed.

---

## [Decision Letter · Decision Letter 2]

20 Mar 2023

Distinctive features of lipoprotein profiles in stroke patients

PONE-D-22-24906R2

Dear Dr. Konishi,

We’re pleased to inform you that your manuscript has been judged scientifically suitable for publication and will be formally accepted for publication once it meets all outstanding technical requirements.

Kind regards,

Gulali Aktas

Academic Editor

PLOS ONE

Additional Editor Comments (optional):

Required revisions are followed. The paper is improved significantly.

Reviewers' comments:

Reviewer's Responses to Questions

**Comments to the Author**

1. If the authors have adequately addressed your comments raised in a previous round of review and you feel that this manuscript is now acceptable for publication, you may indicate that here to bypass the “Comments to the Author” section, enter your conflict of interest statement in the “Confidential to Editor” section, and submit your "Accept" recommendation.

Reviewer #2: All comments have been addressed

2. Is the manuscript technically sound, and do the data support the conclusions?

Reviewer #2: Yes

3. Has the statistical analysis been performed appropriately and rigorously? 

Reviewer #2: Yes

4. Have the authors made all data underlying the findings in their manuscript fully available?

Reviewer #2: Yes

5. Is the manuscript presented in an intelligible fashion and written in standard English?

Reviewer #2: Yes

6. Review Comments to the Author

Reviewer #2: (No Response)

7. PLOS authors have the option to publish the peer review history of their article (what does this mean?). If published, this will include your full peer review and any attached files.

Reviewer #2: **Yes: **Francesco Di Lorenzo

---

## [Editor Report · Acceptance letter]

28 Mar 2023

PONE-D-22-24906R2 

Distinctive features of lipoprotein profiles in stroke patients 

Dear Dr. Konishi:

I'm pleased to inform you that your manuscript has been deemed suitable for publication in PLOS ONE. Congratulations! Your manuscript is now with our production department. 

Kind regards, 

on behalf of

Professor Gulali Aktas 

Academic Editor

PLOS ONE